# Spatializing stigma-power: Mental health impacts of spatial stigma in a legally-excluded settlement in Mumbai, India

**Saloni Dev**[1,2,3]*, **Jasper Duval**[4], **Amith Galivanche**[3], **Tejal Shitole**[5], **Kiran Sawant**[5], **Shrutika Shitole**[5], **Anita Patil-Deshmukh**[5], **Alisa Lincoln**[2], **Ramnath Subbaraman**[3,5,6‡], **Liza Weinstein**[4‡]

**1** Department of Health Sciences, Bouvé College of Health Sciences, Northeastern University, Boston, MA, United States of America, **2** Institute for Health Equity and Social Justice Research, Northeastern University, Boston, MA, United States of America, **3** Department of Public Health and Community Medicine, Tufts University School of Medicine, Boston, MA, United States of America, **4** Department of Sociology and Anthropology, College of Social Sciences and Humanities, Northeastern University, Boston, MA, United States of America, **5** PUKAR (Partners of Urban Knowledge, Action and Research), Mumbai, MH, India, **6** Division of Geographic Medicine and Infectious Diseases, Boston, MA, United States of America

‡ RS and LW are joint last authors on this work.
* saloni.dev@tufts.edu

**Data Availability Statement:** Given confidentiality and privacy concerns, de-identified qualitative data will be available from authors upon request. These data were collected through in-depth interviews

## Abstract

In disadvantaged neighborhoods such as informal settlements (or "slums" in the Indian context), infrastructural deficits and social conditions have been associated with residents' poor mental health. Within social determinants of health framework, spatial stigma, or negative portrayal and stereotyping of particular neighborhoods, has been identified as a contributor to health deficits, but remains under-examined in public health research and may adversely impact the mental health of slum residents through pathways including disinvestment in infrastructure, internalization, weakened community relations, and discrimination. Based on analyses of individual interviews (n = 40) and focus groups (n = 6) in Kaula Bandar (KB), an informal settlement in Mumbai with a previously described high rate of probable common mental disorders (CMD), this study investigates the association between spatial stigma and mental health. The findings suggest that KB's high rate of CMDs stems, in part, from residents' internalization of spatial stigma, which negatively impacts their self-perceptions and community relations. Employing the concept of stigma-power, this study also reveals that spatial stigma in KB is produced through willful government neglect and disinvestment, including the denial of basic services (e.g., water and sanitation infrastructure, solid waste removal). These findings expand the scope of stigma-power from an individual-level to a community-level process by revealing its enactment through the actions (and inactions) of bureaucratic agencies. This study provides empirical evidence for the mental health impacts of spatial stigma and contributes to understanding a key symbolic pathway by which living in a disadvantaged neighborhood may adversely affect health.

with individuals living in marginalized urban slum communities and audio recordings were transcribed into detailed narrative transcripts. We only present de-identified data in the manuscript, with study participants' names anonymized using pseudonyms, and we were very careful to provide limited information from the interview transcripts, such that there is no risk that study participants could be identified. Our concern with public deposition of the full narrative information is that, even with de-identified transcripts, these extended interview data provide enough specific information that there is a risk that study participants could be identified. As such, release of this narrative data could result in the risk of ethical breaches, particularly breaches of confidentiality. Our informed consent forms specifically noted that only excerpts of information would be provided in published manuscripts or other research outputs, such that individuals would not be identifiable. As such, PUKAR's Institutional Ethics Committee imposes such restrictions upon the data. Requests for secure access to the data could be sent to Mr. Shahazade Akhtar (Administration Lead at PUKAR) at pukar@pukar.org which will be then reviewed by PUKAR's Institutional Ethics Committee (FWA00016911).

**Funding:** The study conceptualization, study design, hiring of the study team, data collection, and data transcription and translation were supported by a grant from the National Institutes of Health Fogarty International Center (R24 TW007988; awarded to RS as a fellowship, including salary support for RS). The Rockefeller Foundation provided additional support awarded to PUKAR, with APD as one of the principal investigators. This grant included salary support for APD, TS, KS, SS, and RS. Data analysis and interpretation were partly supported by grants from the National Science Foundation (#1918175 (awarded to RS, including salary support for RS and AG) and #1918128 (awarded to LW, including salary support for SD and JD)). The funders had no role in the study design, data collection and analysis, decision to publish, or preparation of the manuscript.

**Competing interests:** The authors have declared that no competing interests exist.

# Introduction

Geographic inequalities in health have been previously noted globally with disadvantaged, low-income neighborhoods experiencing the worst health outcomes compared to advantaged and affluent neighborhoods. Such inequalities have been attributed to factors operating at multiple levels. At the meso-level, infrastructural factors include poor living conditions and a lack of availability of healthy foods and green spaces [1–3], as well as social factors, such as concentrated poverty and neighborhood crime level [4–6]. Moving upstream to the macro-level, distal drivers of neighborhood disadvantage include redlining and the legal status of neighborhoods [7–9]. Underlying this are acts of systemic discrimination that impact the uneven distribution of resources, education, and employment opportunities within and between localities [10, 11]. This discrimination often stems from the stigma attached to particular geographic locations, which creates perceived distinctions between 'acceptable' and 'unacceptable' neighborhoods, and between 'deserving' and 'undeserving' residents. Although systemic discrimination has been described extensively in studies of health disparities, the question of how spatial stigma shapes residents' subjective understandings, lived experiences, and thereby their health and well-being, remains underexplored in public health research [12].

Even as increased attention has been paid to the impacts of systemic discrimination on population health, spatial stigma has been overlooked by researchers and remains underrepresented in public health debates. In the past decade, only a few studies have examined how residents' subjective experiences of spatial stigma shape their health in under-resourced, stigmatized neighborhoods (e.g., Garthwaite & Bambra [35]; Tabuchi et al. [34]; Wutich et al. [28]). To reduce geographical health disparities, there is a timely need to understand and address spatial stigma, as it stems out of systemic discrimination, and how it shapes residents' lived experiences, in addition to improving neighborhoods' material conditions.

In this paper, we investigate the pathways through which spatial stigma impacts the mental health of residents in a non-notified slum (i.e., settlement lacking legal recognition by the government) in Mumbai, India.

# Literature review

Previous studies have linked neighborhood disadvantage characterized by poor physical and social environments to worse health outcomes [13–16]. Such disadvantage is characteristic of informal settlements or "slums" which have been defined by the United Nations Human Settlement Program (UN-Habitat) as "lacking one or more of the following indicators: a durable housing structure; access to clean water; access to improved sanitation; sufficient living space; and secure tenure" [17]. In general, slum populations suffer from health outcomes that are poorer than those of formally housed urban populations and lower-income rural populations [18, 19]. Public health research in slums has gained traction in the past two decades with the United Nations Millennium goal to achieve "significant improvement in the lives of at least 100 million slum dwellers by the year 2020," [20], an emphasis that has continued in the Sustainable Development Goals era.

As with other research at the intersection of health and place, the structural determinants of health, including environmental and social factors, have been given particular attention in slum health studies. Physical characteristics of slums such as overcrowding, poor environmental conditions, and inadequate supply of basic water, sanitation, and hygiene (WASH) facilities, as well as social conditions like concentrated poverty, tenure insecurity, and limited access to healthy foods, have been identified as determinants of poor physical and mental health among slum residents [21–28]. In India, which has one of the largest populations of slum residents, deprivation in access to basic services and unequal access to opportunities in slums have

been found to be associated with psychological distress and probable CMDs, including depression, anxiety, and suicidality [24, 29, 30]. A previous study in a slum in Mumbai reported that 23% of individuals had a probable CMD, which was the highest rate described in any population-based study in India [24]. Beyond these material and social conditions, the discrimination experienced by many slum residents has also been associated with poor mental health, particularly for women [29, 31]. Although limited research exists on the causal pathways underlying these associations, we hypothesize that these social conditions and experiences lead to chronic stress, which has been shown to be strongly associated with future risk of developing CMDs [32]. However, by shaping subjective experiences of neighborhood disadvantage and associated status loss, spatial stigma may be an important and under-examined pathway by which these structural deficits and discrimination may lead to poorer health outcomes in slums.

Loïc Wacquant [33] first suggested that "blemish" of place can add another layer to understandings of neighborhood disadvantage. Subsequently, urban sociologists and geographers have further developed the concepts of "territorial stigma", "spatial stigma," or "place-based stigma" (for uniformity, we use the term "spatial stigma") to describe the negative perceptions that non-residents have toward disadvantaged neighborhoods. However, only a few public health studies have moved from identifying the physical characteristics of place to examining how the perceptions and subjective understandings of place may impact residents' physical and mental health (e.g., Tabuchi et al. [34] (in Japan); Garthwaite & Bambra [35] (in Northeast England); Kelaher et al. [36] (in Australia); Thomas [37] (in Wales)). Limited research on spatial stigma and health has been conducted in South Asia. However, an exception to this is a recent study based in Dhaka, Bangladesh [38], that reported internalization of spatial stigma among residents of informal settlements along with a tendency to resist such stigma by producing counter-narratives that portrayed their settlement as a "good place". Halliday et al. [12] has recently referred to spatial stigma as "the elephant in the room" for public health researchers, given the limited focus on places' negative perceptions, relative to their likely public health impacts.

In an earlier comprehensive review of studies in this area, Jamie Pearce [39] summarized five non-mutually exclusive pathways that appear to underlie the association between spatial stigma and poor health outcomes (see, Fig 1 for our visualized interpretation of these

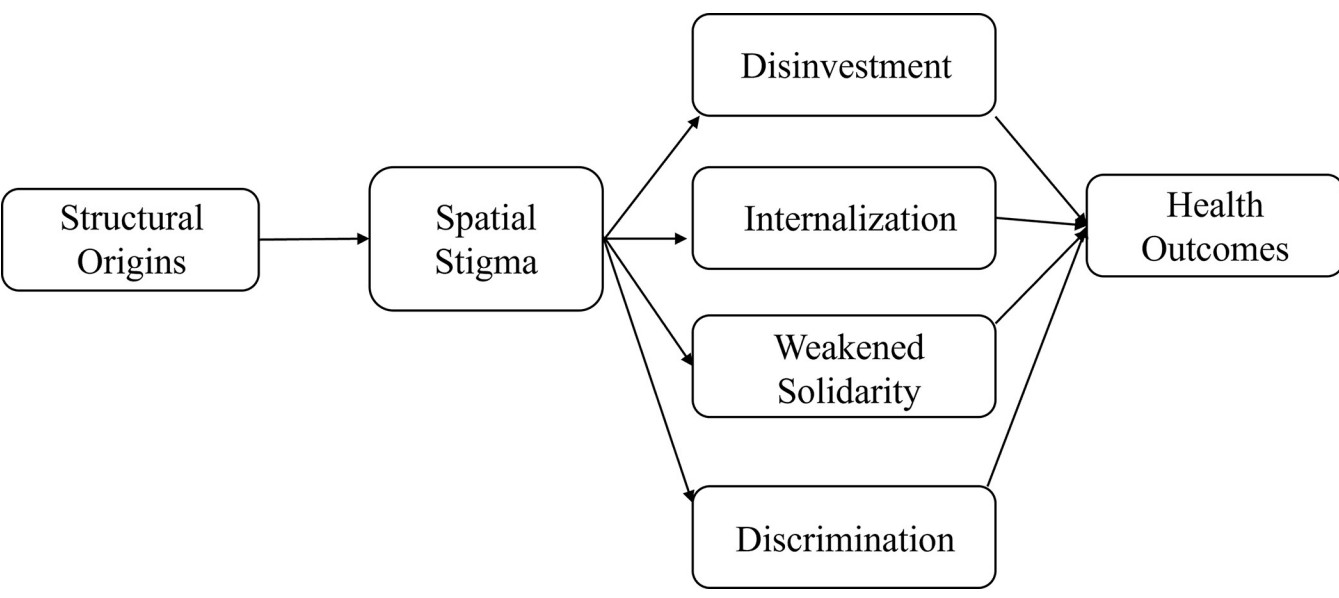

**Fig 1. Pathways underlying the association between spatial stigma and health as summarized by Pearce [39].**

pathways). The first pathway in the review identifies the structural forces that produce spatial stigma in the first place, which are particularly relevant to historically marginalized neighborhoods such as slums. Drawing on Wacquant's [33] original study of spatial stigma, Pearce posits that socio-economic and political processes over time produce a negative perception of the place and attach those perceptions to its residents. In a more recent study, Dana Kornberg [40] has made explicit the so-called "structural origins of territorial stigma" through a detailed empirical study of municipal water allocation in Detroit.

Second, Pearce [39] articulates that stigmatized places tend to suffer from disinvestments in social and physical infrastructures, including housing and access to basic services which are critical social determinants of health. Third, residents may internalize the negative perceptions associated with their place of residence impacting their identity and self-esteem, which can further lead to feelings of embarrassment and shame, as well as behaviors such as concealment, that can negatively impact physical and mental health. Fourth, spatial stigma may negatively affect residents' social relationships and place-based social networks, which may sever social bonds and blunt collective efficacy, thereby further undermining mental health. Fifth, spatial stigma may translate into enacted stigma whereby residents of a stigmatized place are looked down upon and discriminated against, leading to lack of access to educational, employment, and social opportunities, which are also important social determinants of health. It is important to note that Pearce [39] acknowledged that these pathways are non-mutually exclusive but did not articulate the relationships between them.

Recent theories from the stigma literature, some of which were published after Pearce [39], may help shed further light on these pathways. First, Pearce's [39] emphasis on the sources of stigma aligns closely with the concept of "stigma-power" developed by Link and Phelan [41]. Stigma power draws attention to the enactment of stigma through the exploitation, exclusion, or exertion of control over stigmatized individuals. Link and Phelan [41] acknowledge that "structural discrimination disadvantages stigmatized groups cumulatively over time via social policy, laws, institutional practices, or negative attitudinal social contexts." However, most applications of stigma-power have emphasized the interpersonal enactment of power by individual stigmatizers rather than its collective enactment by institutions. Yet when 'spatialized,' or analyzed at a neighborhood or community level, we posit that a focus on stigma-power can help reveal the structural mechanisms, such as the role of bureaucratic agencies, that actively produce the neighborhood's negative perceptions. By focusing on the exercise of power, the concept of stigma-power can help de-naturalize the stigma attached to disadvantaged places and reveal the motives and interests underlying its enactment. Second is the process of "lateral denigration" proposed by Wacquant. Lateral denigration is the process through which residents of a stigmatized neighborhood actively re-direct stigma towards their neighbors, to distance themselves from the spatial stigma associated with their neighborhood. This is crucial especially if we consider the pathway through which spatial stigma impacts health by deteriorating community relationships. Lateral denigration may be seen as a way to cope with self-stigma. Third, Corrigan and colleagues [42] introduced the "why try" effect whereby the effects of self-stigma (including a decrease in self-efficacy and self-esteem) can reduce people's likelihood of participating in activities that can improve their conditions, such as employment or collective action against social and legal exclusion.

These processes may serve as key links to further understanding the impact of spatial stigma on mental health, especially in slums. In the analysis that follows, we use this conceptual framework for interrogating pathways between spatial stigma and mental health in a slum settlement in Mumbai, India, that has been previously shown to have a very high prevalence of CMDs [24]. We also examine the applicability and relevance of Pearce's [39] conceptual framework

in the slum context in India, and the inter-relationships between these pathways. We begin this investigation by providing an overview of the history of our research setting.

## The structural origins of spatial stigma in Kaula Bandar

The slum settlement of Kaula Bandar (KB) is located on land belonging to the Mumbai Port Trust (MbPT, formerly called Bombay Port Trust), an autonomous corporation overseen by the Government of India, at the central or federal level. An examination of KB's history reveals multiple dimensions of systemic discrimination that have shaped perceptions of the community as a stigmatized space. This includes the relegation of port lands to Mumbai's physical and economic periphery, the legal status of the land on which it is located, and the ethnic and religious composition of migration to the community.

KB's history dates back to the 19th century, when it was then called "Tank Bunder" and operated as a dock to load and unload cargo ships. As international trade increased during the height of India's colonial development, private agencies and shipping companies were permitted to build docks in Bombay (Mumbai was called Bombay under the British Colonial Government). The construction of Tank Bunder and other *bunders* (or docks) along Bombay's eastern port led to an exponential increase in trade and the need for a central agency to oversee shipping activities. In the late 1800s, the Bombay Port Trust was established and given ownership of the privately-operated docks, including Tank Bunder [43] (see, Fig 2).

After India's independence from Great Britain in the late-1940s, the Bombay Port Trust came under the authority of the Government of India. In addition to the movement of legally regulated commercial goods, Tank Bunder also became a site where illicit goods, such as liquor, cash, gold, and electronic consumer goods, arrived to avoid detection and taxation. Around this time, Tank Bunder came to be referred to as Kaula Bandar (KB), in reference to

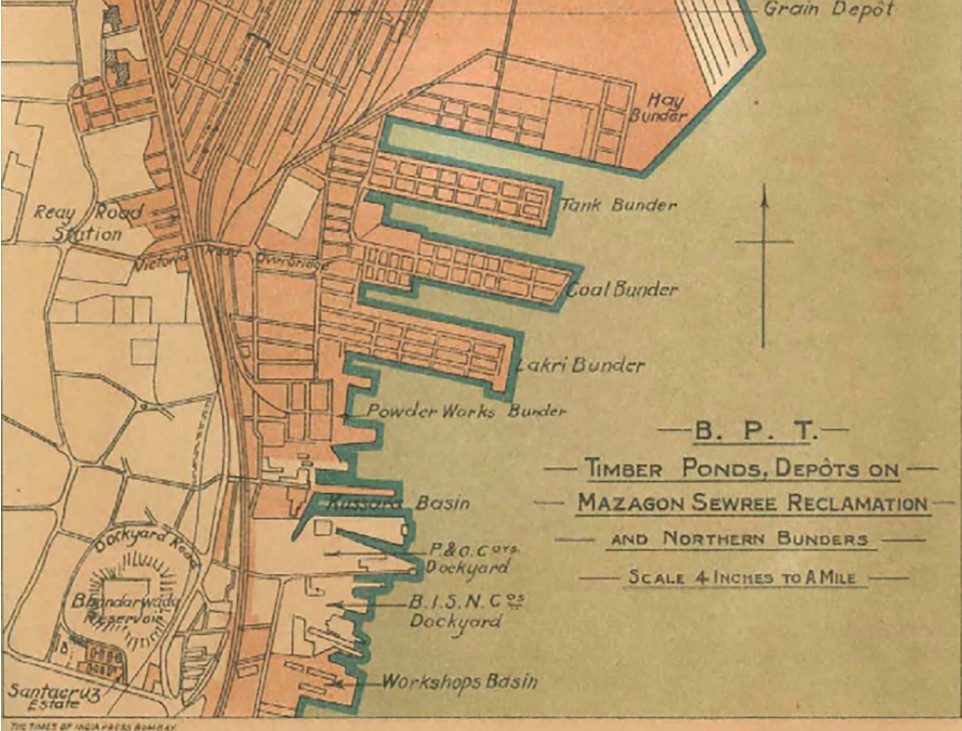

**Fig 2. 1930 map of Bombay's eastern waterfront and Tank/Kaula Bandar [44].**

clay tiles (or *kaula*) that were frequently unloaded at the dock in this period. In the early 1970s, people in need of housing began to construct makeshift homes on KB and on adjacent shipping docks, while bus routes, basic infrastructure, and post offices were established along the eastern waterfront. In the mid-1980s, a new modern shipping port was constructed across the harbor in Navi Mumbai. As high-volume shipping activities moved out of KB, the area was further neglected by local authorities.

The growth of KB's residential population in the early 1970s coincided with a wave of migration to Mumbai from the South Indian state of Tamil Nadu. The growth of Mumbai's Tamil population during this period resulted in considerable ethnic scapegoating, particularly by members of the ethnic-nationalist Shiv Sena party. Additionally, given a small number of high-profile organized criminal actors of Tamil origin in Mumbai, the city's Tamil population was increasingly stereotyped as criminals. Since the 1990s, KB has also seen a growth of North Indian migrants, who now comprise a majority of the neighborhood's population. Many North Indian migrants belong to religious (e.g., Muslim) or regional (e.g., Bihari) identities that have also been marginalized by the Shiv Sena party and other ethno-nationalist political movements.

Because the land of KB is owned by the central government, under the administration of the MbPT, state-level protections and provision of basic services available to residents of 'notified' slums (i.e., settlements legally recognized by the government) are not extended to KB residents. Furthermore, other government agencies responsible for provision of basic services, including the Municipal Corporation of Greater Mumbai (MCGM), the city's municipal government, have also disinvested from the area [45]. In line with recent research identifying "state institutions as sites for the propagation of stigma" [40], we find that deficits in basic services and land tenure supports, arising in part from KB's non-notified status, have contributed to the poor perceptions and stigmatization of the neighborhood. This also supports Pearce's [39] contention that structural factors are foundational to origins of spatial stigma.

## Methods

### Ethics statement

The study protocol was approved by PUKAR's Institutional Ethics Committee (FWA00016911) in February 2011. Verbal informed consent was collected from participants using a script approved by the ethics committee by PUKAR-based researchers (authors KS, TS, SS, RS, and APD). The researchers read a scripted consent form that described the purpose of study, emphasizing voluntary participation and withdrawal from the interviews/FGDs at any time. Participants were given time to ask questions, after which, if they agreed to participate, the researcher signed the form and recorded participant's name to indicate consent. A copy of the consent form was provided to all participants and included contact information for principal investigators at PUKAR. Participants were also encouraged to contact PUKAR with any concerns.

### Study setting

Our study utilizes data collected in Kaula Bandar (KB), which has a population of about 12,700 people (as of 2020). KB has poorer health and educational outcomes when compared to other, primarily notified slums in Mumbai, due in part to the lack of legal mechanisms to provide basic services such as water, electricity, and sanitation to slums lacking recognition [8]. Nearly one-quarter of randomly sampled KB residents in a previous study had a probable CMD, which was one of the highest rates described in Indian population-based surveys [24]. In addition, in that study, exposure to slum-related environmental and social stressors was strongly

associated with the risk of having a probable CMD, independent of income. Additional studies in KB have emphasized the specific role of stress related to poor water access and high water costs in compromising mental health, quality of life, and community social cohesion [25, 46].

The current study expands the scope of this prior work beyond its primary focus on infrastructural deficits and other material causes of ill-health, and considers the subjective perceptions associated with the community's legal exclusion. This study analyzes qualitative data initially collected to inform the development of a slum adversity index in KB and investigates how spatial stigma may contribute to the earlier documented high rates of probable CMD among KB residents. As such, a key premise of this study is that the slum adversities described previously operate at subjective level—rather than operating at a material level alone—to influence the mental health of community residents.

Data were collected by Partners for Urban Knowledge, Action, and Research (PUKAR), an independent research collective based in Mumbai that engages in cross-disciplinary, community based participatory action research. Five researchers conducted individual interviews and six focus group discussions (FGDs) in November 2011. For the interviews, $n = 40$ participants were purposely sampled and interviewed in early 2012 to ensure representation of major ethnic and religious groups in KB (Table 1). Each FGD included 6 to 9 participants. Three FGDs were composed of only women and included 25 participants with ages ranging from 22 to 65

**Table 1. Demographic composition of individual interview participants ($n = 40$).**

|  | Count | % |
| --- | --- | --- |
| **Gender** | | |
| Men | 19 | 48% |
| Women | 20 | 50% |
| Transgender | 1 | 2% |
| **Ethnicity*** | | |
| North Indians | 17 | 43% |
| South Indians | 17 | 43% |
| Maharashtrians | 6 | 15% |
| **Religion** | | |
| Muslims | 20 | 50% |
| Hindus | 15 | 38% |
| Christians | 3 | 8% |
| Buddhists | 2 | 5% |
| **Education** | | |
| Some education | 26 | 65% |
| Never enrolled in school | 14 | 35% |
|  | **Median** | **Range** |
| **Age** | | |
| Men | 38 years | 25–72 |
| Women | 33.5 years | 21–55 |
| Transgender | 48 years | - |

*"North Indians" are usually natives or descendants of natives of the northern states of Delhi, Jammu and Kashmir, Himachal Pradesh, Punjab, Uttaranchal, and Haryana. "South Indians" are natives or descendants of natives of the southern states of Tamil Nadu, Karnataka, Kerala, Andhra Pradesh, and Telangana. We characterize participants based on these three ethnic identities given the history of ethnic scapegoating against North and South Indians in Maharashtra, as well as the ethno-nationalist movement promoting "Maharashtrians" or natives/descendants of natives of Maharashtra (as described in the section "Structural Origins of Spatial Stigma in Kaula Bandar").

years. Three FGDs, composed of only men, included 21 participants with ages ranging from 19 to 70 years. All FGDs were conducted in Hindi, while individual interviews were conducted in Hindi, Marathi, or Tamil. FGDs and interviews lasted 40 to 60 minutes and were audio recorded. Data from the audio recordings was transcribed directly to English.

We emphasize that, although the data analyzed for the current study were collected in 2011 and 2012, these data remain highly relevant given that KB continues to be non-notified. Hence, the underlying structural challenges contributing to the community's marginalization have not changed over the last decade. In addition, we aim to make a contribution to public health and sociological theory, for which the time period when the data were collected is less relevant than the extent to which the data facilitate exploration of relevant concepts.

The purpose of the qualitative guide, which was similar for individual interviews and FGDs, was to identify life adversities or stressors that are characteristic of, or unique to, the physical and social environment in non-notified Indian slums. Initial open-ended questions allowed participants to describe adversities, stressors, or traumatic events experienced at the individual, family, and community levels, based on a socio-ecological framework [47]. Participants were then asked semi-structured questions that prompted them to describe their experiences with specific adversities—such as overcrowding, inadequate structural housing quality or location, and lack of sanitation or water access—that comprise the UN Habitat definition of a slum [48]. A list of open-ended questions from the interview guide can be found in Table 2.

After enumerating adversities, participants were then encouraged to narrate parts of their life history relevant to each adversity or traumatic event, which allowed us to capture the rich context of individual experiences. Our interview guide did not directly ask questions about stigma or mental health, so as not to lead participants towards an overly narrow framing of their life and community adversities. However, perhaps not surprisingly, in relating their experiences of adversity or trauma, participants frequently described how these challenges directly affected their mental state, including feelings of psychological distress. Across nearly all participant narratives, at least some of each person's life adversities were strongly shaped by the community's physical location and legal status. The current study therefore analyzes these narratives with the specific goal of unpacking the relationship between spatial stigma and mental health, both of which emerged as the natural fallout of detailed interviews examining adversities, stressors, and traumatic events in this marginalized community.

For the current paper, the data were analyzed using a deductive approach whereby the pathways underlying the association between spatial stigma and health outlined by Pearce [39] were developed into a thematic framework. Pearce's framework especially lends itself to analysis of the data guided by the socio-ecological framework given that Pearce's pathways assume a multi-level approach to spatial stigma ranging from structural origins of spatial stigma and resultant disinvestment at the macro-level, discrimination and weakened solidarity at the

**Table 2. Open-ended questions from the qualitative interview guide.**

| |
|---|
| 1. What major adversities do *you* currently face on a day-to-day basis from living in your community? Your family? Your community? |
| 2. What major adversities have *you* faced in the *past* due to living in your community? Your family? Your community? |
| 3. What major traumatic experiences have you faced recently or in the past? |
| 4. For each adversity or traumatic experience you mentioned, can you share a story or experience of a time that you faced this problem? |
| 5. You have spoken with me about several different adversities and traumatic experiences that you have faced from living in Kaula Bandar. How do you cope with these problems? Are there people or groups who help you to deal with these adversities? |

community and inter-personal levels, and internalization at the individual level. As such, our use of the socio-ecological framework in our FGD and interview guides to explore challenges faced by participants at the individual, family, and community levels is what enabled us to map findings onto the multi-level processes that shape spatial stigma in Pearce's framework.

Transcripts were coded using Dedoose software (V.8.0.35, Los Angeles, California: Socio-Cultural Research Consultants, LLC). Each transcript was independently coded by at least two members of the data analysis team, which was comprised of three coders (SD, JD, and AG). Coding discrepancies were resolved through discussion and consensus was reached on each transcript. Throughout this process, the team identified representative quotes to illustrate the relevant pathways. Frequent meetings of the coding team with senior authors and collaborators fostered a reflexive analysis process and refined understanding of individual and social dynamics. We report results from our qualitative data analysis using pseudonyms that are reflective of participants' age, gender, and ethnicity.

## Researcher positionality and authorship

Before we present our findings, we would like to discuss our reflections on our position as researchers and authors of this manuscript. The first author (SD) is an India-born scholar with expertise in global mental health. She spent the formative years of her education and training in India before moving to the United States (US) for graduate school. She conceptualized the current study, led the data analysis efforts, and wrote the first draft of this manuscript along with subsequent revisions. The second and third authors (JD and AG) are US-based undergraduate students (at the time of manuscript writing) with training in public health, sociology, and qualitative research methods. They both contributed to the analysis of the data for this study. TS, KS, and SS are PUKAR-based field researchers (at the time of data collection). They were all born and brought up in Mumbai, and have been trained in research methods as a part of PUKAR's "Youth Fellowship" program. Along with APD and RS, they initially collected the qualitative data that were re-analyzed for the current study. They also contributed towards interpretation of the findings. APD is an India-born public health researcher who was trained as a neonatologist in the US. Since her relocation to India in 2005, as the Director of PUKAR, she has focused her research efforts on urban poverty, social determinants of health, and urban knowledge production through the lens of marginalized youth. In addition to data collection, APD contributed towards interpretation of the data and manuscript writing. AL is a US-based public mental health researcher with expertise in mental health stigma. She contributed towards data analysis, interpretation, and manuscript writing. RS is an Indian-American U.S.-based infectious diseases physician and an epidemiologist with expertise in slum research. For more than a decade now, he has collaborated with PUKAR to conduct research on social determinants of health in slums in India, including two years living in Mumbai and conducting field research on a weekly basis in KB. RS conceptualized, designed, and obtained funding, and obtained ethical approvals for the original mental health study under which the data for this study were collected. He led data collection and contributed to data interpretation and manuscript revisions. LW is a US-based sociologist and has dedicated her career to researching slums in Indian cities. She supervised data analysis, and also contributed towards data interpretation and manuscript revisions, based on input from all the authors. All authors provided critical feedback and helped shape the research, analysis, and manuscript writing. The authorship sequence reflects the secondary nature (i.e., re-analysis of data) of the study which entailed re-visiting the data from a new perspective of a different theoretical model. We acknowledge though that the background of the data analysis team may have influenced the initial interpretation of the data. To avoid bias, multiple meetings were held to discuss interpretations and

any preconceptions, and multiple rounds of revisions were made to the manuscript, based on inputs of all authors, to accurately reflect the study findings.

## Results

Results are presented according to the five pathways shaping spatial stigma summarized by Pearce [39] (Fig 1). Specifically, we describe the structural origins of spatial stigma, disinvestment in infrastructure, internalization of spatial stigma, spatial stigma and weakened solidarity, and experiences of discrimination leading to limited life-chances. Fig 3 is an adapted version of Fig 1 highlighting the unique contributions of our results to Pearce's [39] conceptual model, especially around inter-relationships between these pathways (in red dotted lines), and the association with mental health. Most importantly, we found that structural factors independently influenced disinvestment in infrastructural resources for KB. Our analysis reveals that such disinvestment reflects the exercise of stigma power to achieve the goals of eviction and clearing up the land on which KB sits. In addition, our participants reported internalizing spatial stigma enacted through disinvestment and discrimination against them. Lastly, internalization of spatial stigma also directly impacted community solidarity as residents tried to push stigmatizing perceptions onto their neighbors through the process of lateral denigration. Table 3 provides a summary of the results for each of these pathways.

### Structural origins of spatial stigma and disinvestment in infrastructure

Our analysis revealed an intricate relationship between two of Pearce's [39] pathways concerning structural factors and infrastructural deficits, both of which had an undertone of the concept of "stigma power" [41]. We, hence, present findings along these pathways together in a single section.

Our data suggest that political processes may construct spatial stigma over time. Stigma power was actively utilized by public authorities in the form of social disinvestment and

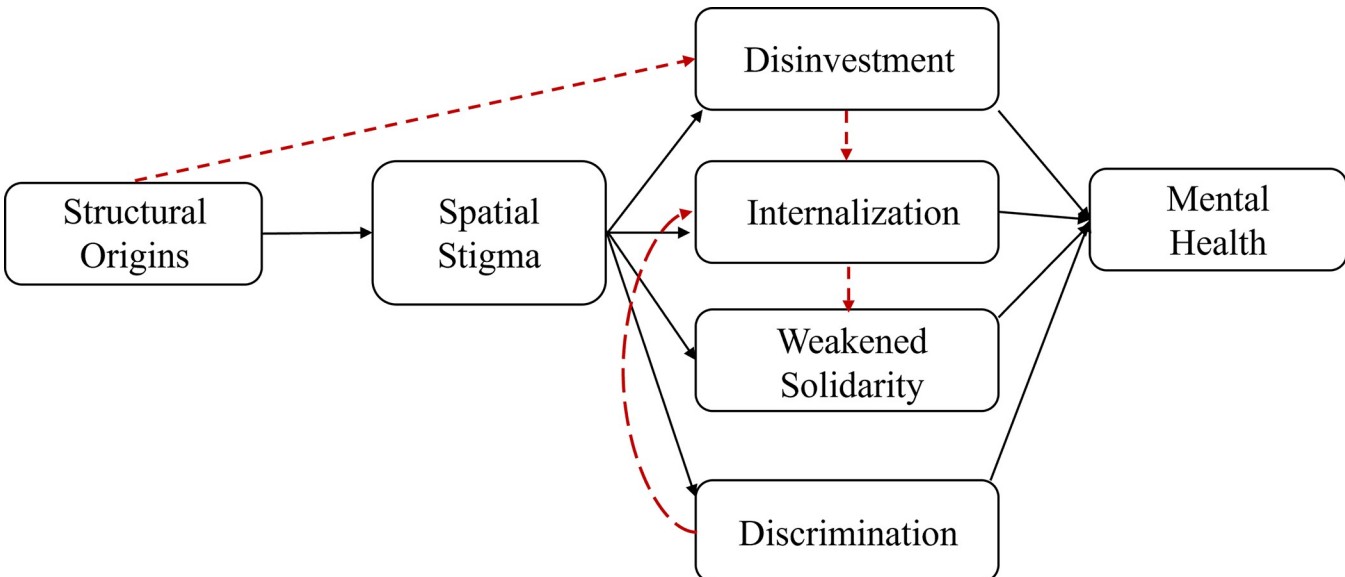

**Fig 3. Adapted Pearce's conceptual model and inter-relationships between pathways.** Black boxes and lines represent Pearce's conceptual model. Red dotted lines represent the unique contribution of our results: structural factors independently led to disinvestment in infrastructural resources. Spatial stigma, enacted through disinvestment and discrimination was a major pathway to internalization of such stigma. Internalization impacted community solidarity signifying via lateral denigration.

**Table 3. Snapshot of findings according to pathways summarized by Pearce's [39].**

| Mechanisms | Attributes | Quotes |
|---|---|---|
| Structural Origins | Use of stigma power with the goal of clearing up land | *"I stopped them and asked them, "Why are you releasing these stray dogs in our area? People live here also! Why are you not thinking about our welfare? We are not animals." He never replied to me."* (Gopalan) |
| Disinvestment | Social disinvestment, infrastructural deficits, and collective punishment | *"The electricity and water problems exist all across Mumbai. Compared to the rest of Mumbai, we do face somewhat greater problems with water and electricity. But the police harass us much more than people in other communities. The police always take bribes from us and from the thieves [. . .] If we want to file a complaint, we go to [a police station in] another area [. . .] There is no safety here [. . .] At nighttime, ladies cannot safely walk on the road."* (Ahmed) |
| Weakened Solidarity | Lateral denigration, and strained community relations. | *"If people want to, then they can clean their area. But no one takes this issue seriously. Everyone throws their garbage into the sea. The BMC car comes here to pick up the garbage. But no one throws their garbage in a place the BMC can pick it up. If they threw their garbage there, then we wouldn't face this problem."* (Akshay) |
| Internalization of Spatial Stigma | Awareness and acceptance of others' negative perceptions, and feeling shameful for their neighborhood. | *"I think all the time that [I am not a part of the city]. We have our name on the voting card. We always go to vote, but look at the condition of our area. It does not look like the rest of Mumbai. I feel like people treat us differently [. . .] They don't consider us a part of Mumbai. We live in garbage."* (Padma) |
| Experiences of Discrimination | Being discriminated against by local administrators, potential employers, police, and residents of other parts of the city. | *"Currently, [I am] not employed. I have been looking for a job, but no one is willing to give me a job once they hear that I am from KB and [that] I am Muslim. This has irritated & frustrated me a lot."* (Shahzad) |

collective punishment, for example by cutting non-formal electricity wires or raiding and removing water pumps to cut off the community's water supply. This further marginalized the community by impacting social structures, for example, by enabling the police and other government officials to extract bribes for any basic services that are accessed by residents, since any such access is technically illegal.

Some residents believed that public authorities are using stigma power with the goal of eventually displacing them from their homes to clear the land for use by the government or private actors. Logambal, a 32-year-old Tamilian Hindu woman who has lived her whole life in KB, expressed this sentiment while describing challenges in accessing water after the periodic community-level raids by MbPT to confiscate "illegal" water pumps all at once:

> *"[This causes] unbearable tension. "[I feel like they do such raids across KB periodically] because we are on BPT land. I feel like they are stopping the water because they eventually want us to leave this land."*

Others residents reported that some industries were storing coal close to KB, even after the residents voiced their concerns about its harmful effects on their health. In other instances, residents reported that the MCGM would occasionally release stray dogs caught in other areas in Mumbai in KB. A 36-year-old Tamil Buddhist and long-time KB resident, Gopalan, explained that he once encountered officials from the BMC releasing stray dogs in KB and confronted them.

> *"I stopped them and asked them, "Why are you releasing these stray dogs in our area? People live here also! Why are you not thinking about our welfare? We are not animals." He never replied to me."*

These practices created spatial stigma which, in turn, further marginalized KB residents through inaction by the public authorities such as the police, a manifestation of the pathway

through which spatial stigma discourages investments in the development of the stigmatized space, thereby undermining mental health.

For example, Ahmed, [25-year-old Muslim man] a KB-resident and a call center employee, explained in a FGD:

> *"The electricity and water problems exist all across Mumbai. Compared to the rest of Mumbai, we do face somewhat greater problems with water and electricity. But the police harass us much more than people in other communities. The police always take bribes from us and from the thieves [. . .] If we want to file a complaint, we go to [a police station in] another area [. . .] There is no safety here [. . .] At nighttime, ladies cannot safely walk on the road."*

He later added:

> *"50 people sometimes take illegal connections from one meter. We would have a good amount of water for this area if we got that water every day. But sometimes, people who live outside this area complain that people in KB steal water [. . .] and they stopped our water. This is their politics, because they don't want people to live here."*

Ahmed's account of access to water and electricity in KB clearly demonstrates the cyclic nature of how structural and infrastructural factors are a cause and consequence of the stigma attached to KB. In the absence of legal ways of accessing these fundamental needs, some KB residents obtain water and electricity through informal means, which can be framed through the stigmatizing language of "illegality". This reveals that when the governmental inaction denies the residents legal ways to access water and electricity, stigma power can further amplify stigma. When someone filed a complaint against these residents, their informal connections were taken away, revealing how spatial stigma can lead to further contribute to material marginalization.

Navigating such infrastructural deficits forced many residents to carry an undue financial burden, thereby pushing their families further into poverty, leading to negative impacts on their mental health, described using the word "tension," a term used across India to indicate psychological distress. For example, the periodic community-level raids of water pumps leading to a "system failure" dramatically increases the water costs paid by KB residents as compared to residents of slums with legal water access [46]. A previous study also reported the high costs of diarrheal illness given the lack of access to sufficient quantities and quality of water in KB [26].

For example, Biju [40-year-old Tamil man] is a carpenter who was born in KB. In a FGD, he elaborated on how, despite working hard to provide enough money for his family and children, he experiences tension due to the high cost of informal basic services:

> *"We have been here many years. We won't leave [. . .] no matter [. . .] whether we get water or not. They can take 30 or 50 rupees for water or the police can take 3000 rupees or 5000 rupees from us. Just to build a house here, the police ask for [a bribe of] 40,000 rupees. What will I give to my children if they take that kind of money from me? How can we be without tension in this situation? No one does anything for us, but we want to stay here."*

Similarly, for Noora, a 40-year-old female tutor with five children, dealing with infrastructural deficits entailed risking her personal assets which led to experiences of tension, as explained in a FGD.

*"At midnight, if we lose our electricity, then it's our job to check if someone is stealing our electricity or not. If I go outside to check the meter, I get scared because people think that I'm the one stealing electricity. And if we go there at nighttime to find out who is stealing our electricity, then people may come and steal things from our home while we are gone. That is another cause of tension."*

### Internalization of spatial stigma: Impacts on identity and community solidarity

Our findings also revealed the inter-relationship between impacts of spatial stigma on residents' identities as well as their social relationships through the process of internalization and lateral denigration.

Residents' experiences of living in KB sometimes reflected internalization of negative external perceptions of the community. We identified two main ways that the respondents *experienced* or *enacted* these feelings: through awareness and acceptance of others' stigmatizing perceptions of KB and feeling shameful of their neighborhood.

Residents recounted how others perceived KB residents as criminal, slovenly, and unwanted residents of Mumbai. Moreover, they shared experiences of such perceptions being enacted upon by others, for example, when they endured contempt in public spaces such as restaurants, where slum dwellers may be viewed as outcasts. Some residents further highlighted their acceptance of such perceptions as true and valid, often viewing themselves as well as KB through such a negative lens.

One such account is eloquently elaborated by Padma. Padma, a married home-maker, had faced many challenges while living in KB for 20 years, for example lack of water, limited supply of expensive kerosene, and taking care of her son with developmental disabilities. She wished to move back to her village for better education for her children and a better lifestyle. She noted that KB residents are seldom viewed as citizens of Mumbai by others. Over the years, she has attached these negative views by outsiders to her identity, concluding that she and others live in "garbage".

*"I think all the time that [I am not a part of the city]. We have our name on the voting card. We always go to vote, but look at the condition of our area. It does not look like the rest of Mumbai. I feel like people treat us differently [. . .] They don't consider us a part of Mumbai. We live in garbage."*

In one of the men's FGD, Biju, along with Amir (a 19-year-old Muslim man who has been living in KB since his childhood) and Saia (20-year-old Tamil man who works for the BMC) also discuss such internalization while explaining how community-level crisis (such as a fire) stemming from infrastructural deficits, or inaction by state-actors is inherently characteristic of KB.

*"Aamir: The first time a fire broke out, I was a child. Last summer also, there was a fire. [Some homes that] did not burn in that fire, they were robbed after they were evacuated. The homes are burning in the fire, and people are robbing homes at the same time!*

*Aamir and Sai: The police never took any action against them.*

*Sai: People lost all their documents.*

*Aamir: The police caught innocent people for that robbery.*

*Biju: Our area is like this only."*

KB residents also expressed internalization of spatial stigma when they described feeling ashamed to invite friends or extended family members to their homes owing to the lack of water supply and sanitation in the community. Although these deficits in basic services are the result of government inaction, KB residents personally struggled with feelings of embarrassment and shame, due to their inability to ensure the comfort and safety of their guests.

Furthermore, for Sakina, a 33-year-old Muslim woman and a KB-resident, the conditions in KB, primarily produced by its non-notified status, were a source of embarrassment in front of invited guests. The irony in Sakina's account that follows is that moving to the city from village is often seen as an indicator of economic progress, however, she has opposing perceptions given the conditions in which she lives.

*"We don't give our address to our relatives when we go back to the village. We are afraid that if they come here, they will see the awful conditions that we are living in. We are living here because we don't have money to purchase a home outside of this area."*

While experiences of stigma were broadly shared by residents, the extent to which this stigma was internalized varied among KB residents. For example, Radhika, a resident for 35 years, is the primary caretaker of her household. Although she lacked trust in the police and legal system and experienced psychological distress related to her family, health, and finances, she did not endorse negative perceptions of KB as a community. Instead, she acknowledged that residing in KB for many years has empowered her with a sense of belonging and citizenship, despite the existence of numerous adversities.

*"Just like we were once citizens of Tamil Nadu, we feel the same way about this place now. We feel like anyone would in their native land. Because we've been here for so many years."*

Our findings further underscored that internalization of spatial undermined community relationships. Residents distanced themselves from "other" residents and claimed that the stigma attached to KB applies to their neighbors but not themselves, a process known as "lateral denigration" [33]. Participants sometimes asserted that "people" living in KB are "not good people," because other community members occasionally fight with each other or engage in activities that are technically illegal to facilitate access to basic services such as water and electricity. In addition, residents also blamed the community's inadequate collective efficacy for the conditions in KB. Lateral denigration sometimes compromised social relationships, an important predictor of mental health [49].

Some specific examples of participants displacing stigma onto their neighbors included perceptions that other KB residents often fight with each other to obtain access to basic services, generalization of individual instances of crime (e.g., robbery) as being indicative that the community more broadly is morally compromised, and concerns that other ethnic groups within the community are not "good people." In some cases, concerns about specific ethnic groups in the community reflected that fact that populations who had been in the community for a longer time (e.g., Tamilians) had secured relative control over informal selling of water and electricity.

For example, after living in KB for a decade, Punitha did not like the community and wanted to move back to her village. She was troubled by her son's drinking problem and unemployment, as well as the high costs of basic needs and healthcare. She communicated her perception that KB residents do not respect each other and were always fighting with each other.

*"I tell my son all the time to sell everything and return to our village, but he doesn't listen to me.. . . People who live here don't respect each other. They are always fighting with one another."*

Akshay, a 35 year old man, had been living in KB for two years after migrating from nearby Madanpura. He relayed fears about his children in KB, especially regarding safety and schooling, and decided to send them back to his home village. While discussing sanitary conditions in KB within a FGD, he pointed out how the poor conditions in KB could actually be ascribed to others in KB who do not throw trash in designated spaces.

*"If people want to, then they can clean their area. But no one takes this issue seriously. Everyone throws their garbage into the sea. The [MCGM] car comes here to pick up the garbage. But no one throws their garbage in a place the [MCGM] can pick it up. If they threw their garbage there, then we wouldn't face this problem."*

This perception that residents themselves were not cooperating with basic sanitation procedures was at odds with the reality of the neglect of solid waste collection in KB by the government. At the time data were collected, while the municipal corporation was sending garbage trucks to the docks, these trucks only collected garbage from a single dumpster located immediately outside the slum, which was inaccessible to most of KB's population [8].

Participants also described lack of community, solidarity which contributed to a sense of social isolation felt by some residents. These residents underscored a need for collective action, which was lacking, for improving basic services and making KB a better place to live.

For example, Deepak was in his 30s and worked as a laborer in the transportation industry. He shared accounts of somatization of his stress caused by the problems in KB, especially lack of water and corruption by the police. Deepak believed that the community's problems could be overcome if residents came together and took collective action:

*"Everyone only takes care of their own business—so who is thinking about the larger community? [. . .] If we only worked together and cooperated, we could get more things done [. . .] But everyone else in this community doesn't feel the same way. They don't feel the need to take care of each other. They are only concerned about their own lives. [. . .] It's only if we work together as one, then all of our work will get done. [. . .] If we all just complain about each other all the time, then our work won't get done."*

Narsimha, a 36 year old dockworker and lifelong KB resident, expressed concerns about KB's legality and political corruption. In one of the FGDs, he echoed Deepak's sentiments in terms of access to basic services:

*"In six months, we have another election. We know [the politicians] will start to come here now. They will make promises that we will do this and that. But we know, after the election, they won't do anything. It's our fault also, because we never ask them any questions. Why? Because we don't have any unity. [. . .] If we came together, then they wouldn't have any other option. They would have to listen to us."*

Interestingly, the lack of solidarity was attributed to the influx of residents in KB, leading to diverse interests and ways of thinking. Biju in the FGD explained,

*"Biju: Everyone thinks differently. That's why we don't have any unity. That's why we suffer from lack of water. In the old days, we got water very easily, but now the population of KB has increased. That is why we have problems with water now."*

On the other hand, some residents portrayed a positive picture of community cohesion within KB. Some residents took pride in how, during the 1992–1993 riots in Mumbai following the demolition of the Babri Mosque, KB was not a site of violence. While contrasting his experience of striving to take care of his family and not getting enough support from them, Bilal said,

*"The good thing about this area is that we don't have any big racial tension. People have personal problems, but there are not large groups fighting."*

His account is reflective of how even in the presence of personal problems, KB residents lived peacefully with each other, if not collaboratively. Further analysis revealed that such pride was not an anomaly among KB residents.

Rahman, too, actively disassociated him from external stigma against KB. He was 72 years old and had lived in KB for over 40 years. While he acknowledged difficulties with access to basic services and crime in KB, he viewed these challenges as being pervasive in every neighborhood. He placed blame on higher powers for the poverty and economic inequality experienced by the community:

*"Everyone says KB is not a good place, but I think it is a good place. I know all of Mumbai, I've walked through each and every lane [. . .] Some bad things happen in our area, but it doesn't mean it's bad to live here. Bad things happen everywhere. It all depends on us. If you are good to others then they will behave well back to you. So I don't think that it depends on religion. [. . .] Here, the worst thing is poverty. Because we are poor we face these problems. But it is true that if poor people were not here then it would be difficult for rich people to survive, because they also depend on us. God created this kind of inequality."*

Rather than denigrating other community residents, such counter-narratives often emphasized the role that infrastructural deficits—and therefore external forces such as the government—played in undermining community cohesion. Within the men's FGD, participants reflected on how helpful and supportive their neighbors were, except for when basic services were involved:

*"Interviewer: Your neighbors or relatives, do they not help you?*

*Aamir: Yes, neighbors help each other.*

*Ahmed: They provide electricity from their home. Sometimes they give us water. But they are also poor, so they are not always able to help us.*

*Ehsaan: I asked my neighbor to give water. They said, "If you want food, we are ready to give you food, but don't ask us for water!"*

### Discrimination based on place of residence

Twenty-five participants described KB in terms that suggest the community is discriminated against by local administrators, the police, and residents of other parts of the city. For example, several residents spoke about taxi drivers and others refusing to come to KB. For example, Rahman, a 72-year-old Muslim man, who had lived in KB for 40 years, explained:

*"In the past it was dangerous. Taxi drivers used to not want to come here. See, the entire community is not bad. Only a few people do bad things and make the whole community look bad. In the past, a lot of crime used to happen here. It happens now too, but less often."*

Such enacted stigma was not only limited to denial of transportation but also affected employment and social ties. Shahzad [a 25-year-old Muslim man] first moved to KB as a child. Due to challenges finding a job, he previously engaged in illegal activities such as robbery and feels ashamed for not being able to provide for his family. He described how his vulnerability as a KB resident and Muslim adversely impacted his job opportunities:

*"Currently, [I am] not employed. I have been looking for a job, but no one is willing to give me a job once they hear that I am from KB and [that] I am Muslim. This has irritated and frustrated me a lot."*

In a woman's FGD, participants shared that stigma attached to KB often impacted their social relationships. One of the participants of the FGD, Arthi [28-year-old Tamil], was born in KB and lives with her husband and children. She also works as a sweeper and has completed education until the 5th grade. Another, Gulnaz [40-year-old Muslim] is uneducated and moved to KB after her marriage. She lives with her children and works as a sweeper. Both of them explained,

*"Arthi: When people from our village come here, they don't like this place? They ask us, "Why are you living in this garbage?"*

*Gulnaz: Our guests don't want to stay with us here. They don't like this area."*

## Discussion

Research on place and health has tended to focus on the health effects of physical conditions such as infrastructural deficits, while neglecting the role that perceptions, subjective experiences, and social relations can play in shaping these effects. Responding to the dearth of empirical studies on the mental health effects of spatial stigma, our analysis uses underlying pathways summarized by Pearce [39] to trace spatial stigma attached to KB, a non-notified slum settlement with earlier recorded high levels of probable CMD [24]. Our findings reveal the production of spatial stigma through infrastructural deficits and underlying structural factors. In addition, participants account highlight the stigma power behind the enactment of spatial stigma (i.e., acts of discrimination). Moreover, we traced the impacts of spatial stigma through its recognition and internalization by KB residents, denial of opportunities, strained social relationships at the individual and community level, to its potential impacts on residents' mental health.

Our study is one of the few that has investigated public mental health challenges in a slum in South Asia through a lens of spatial stigma. To the best of our knowledge, the only other study to explore narratives of spatial stigma in South Asia was conducted by Fattah and Walters in Dhaka, Bangladesh, although this study did not have an explicit focus on health. They found that slum residents' narratives simultaneously reflected the internalization of spatial stigma and counter-narratives to produce solidarity. Our findings similarly reveal the simultaneous internalization of spatial stigma and occasional counter-narratives of solidarity; however, we also expand upon this prior work by highlighting other multiple pathways through which spatial stigma impacts resident's mental health, as discussed below. Most importantly,

as compared to Pearce's framework, our findings highlight the centrality of structural factors to creating spatial stigma in slums, as well as potentially in the South Asian context more generally.

Our analysis revealed that, in addition to broader structural forces such as deindustrialization and migration, KB's stigma is produced by government actions. The broad array of actions taken by the government against KB as a community—including raiding of water motors, cutting of electricity wires, harassment of individuals engaged in open defecation, neglect of garbage removal, and release of stray dogs (to name a few)—not only have important material consequences for KB residents, but these actions also have important symbolic dimensions by both enacting stigma and shaping perceptions of KB as a stigmatized space. These findings support the small but growing literature on the role of state actors and agencies in the construction of spatial stigma [33, 40, 50] and provide initial evidence of this phenomenon in Indian slums. Recognizing these state actions as the spatialized enactment of stigma-power, our analysis also helps demonstrate the relevance of the concept of stigma-power for understanding neighborhood-level and interpersonal processes. While the full set of motives underlying the exercise of stigma-power in KB is complex and requires further research, our respondents shed light on land clearance and opportunities for redevelopment as key motivations underlying these actions.

By focusing on the sources of spatial stigma, our analysis also helps reveal a bi-directional relationship between spatial stigma and infrastructural disinvestment, as compared to a uni-directional one suggested by Pearce [39]. For example, without formal or legitimate ways to access to water and electricity, many residents engage in illicit means or are forced to pay bribes to police officers or local administrators, which reinforces perceptions of KB as a site of criminality and corruption. Spatial stigma was not only enacted through disinvestment in infrastructure in KB, but also, in turn, intensified the stigma attached to KB.

In addition, we found initial evidence for the potential impact of spatial stigma on mental health of KB residents. Many residents internalized the negative perceptions attached to KB. These findings align with other studies that have recognized 'self-stigma' as a byproduct of public stigma [51, 52]. Respondents reflecting this perspective reported seeing themselves as living in garbage both literally and figuratively. The effects of self-stigma can include a decrease in self-efficacy and self-esteem, or the "why try" effect [42], which has been reported to be a risk factor for social disorder and mental illnesses [53]. Further, the internalization was also reflected in the embarrassment and stress participants felt in inviting and hosting friends and guests to their homes in KB. These experiences have significant consequences for the mental health of KB residents, as we know quality social relationships can provide emotional and instrumental support, reduce stress, and further contribute to mental health [54, 55].

For other respondents, the internalization of spatial stigma led to lateral denigration, or directing the stigma to other neighborhood residents, while disassociating themselves from the stigmatized identities. Our analysis suggests that these residents utilized this strategy to manage or minimize the personal stress or shame associated with the stigma, but that lateral denigration can also weaken their social ties within the neighborhood which is an important predictor of mental health [49]. Previous research has demonstrated that high social bonding within stigmatized neighborhoods can foster resilience and ability to resist spatial stigma [56]. However, some studies have found the contrary, that strong social networks and interdependencies may be emotionally, physically, or financially burdening, especially for those living in disadvantaged neighborhoods [57]. While the individual-level benefits of neighborhood-based social networks are less clear, community-level conflicts associated with lateral denigration can lead to higher levels of stress and lower levels of collective efficacy. A finding unique to the population composition of KB was the intersection of spatial stigma with ethnic tensions

within KB led to ethnic scapegoating. Residents sometimes directed the spatial stigma and lateral denigration towards the Tamil community. It is imperative to further investigate the interactions between different stigmatized identities within KB and how that shapes the association between spatial stigma and mental health. This may have implications for solidarity and collective action in urban slums and for future implementation of interventions to promote mental health in these settings.

Furthermore, we found evidence for the pathway identified by Pearce [39] that spatial stigma, enacted through discrimination, limits residents access to material resources important for health. The experiences ranged from being denied a taxi ride, to being discriminated against for job opportunities, and being refused a visit to their home by guests. Such took a toll on residents' opportunities to climb the socio-economic ladder as well as maintain healthy social relationships. The impact of these social determinants on mental health has already been shown by numerous studies including in KB [24]. Future studies looking at deprived sections of the society and the important life opportunities available to them could benefit from utilizing the concept of spatial stigma to better understand the underlying mechanisms.

It is important to acknowledge that many respondents actively challenged KB's stigma and attempted to isolate negative characteristics or events from the broader qualities of the neighborhood. Such counternarratives have also been reported by studies on spatial stigma in Bangladesh [38], England [37], and Brazil [58]. As Fattah and Walters [38] explain, such claims do not necessarily present a false perception of neighborhood conditions, but underscore that the lived experiences that do not align with the public perceptions may be overshadowed by spatial stigma. With respect to the effect of counternarratives on mental health, our analysis suggests that further research is needed to understand the role such perceptions and experiences may play in moderating the association between spatial stigma and adverse mental health.

Overall, our findings support the socio-ecological framework that guided the methodology of the original data collection. Specifically, our study highlights the multi-level impacts of social stigma on mental health outcomes. At the individual level, residents of KB internalized spatial stigma, negatively influencing their self-perceptions. Along the same lines, the individual-level pathway identified in the socio-ecological framework emphasizes the importance of psychological and cognitive processes. At the interpersonal level, our study reveals that spatial stigma within KB hampers community relations, and facilitates discrimination against KB residents. This aligns with the corresponding pathway in the socio-ecological framework and underscores the importance of social connections and support. Finally, at the community/macro level, we noted that spatial stigma in KB is perpetuated by willful government neglect and disinvestment, including the denial of basic services like water, sanitation infrastructure, and solid waste removal. The corresponding pathway in the socio-ecological framework illustrates the role of community resources, institutions, and social norms.

## Future directions

We recognize the immediate need for future studies to develop a measurement tool for spatial stigma that would allow for quantitative examinations of its impacts on mental health. A longitudinal analysis, in particular, could help provide evidence for causality and the underlying pathways outlined in this study. In addition, we acknowledge that the data we analyzed are about a decade old; however, KB continues to be a non-notified (i.e., legally excluded) settlement and, because it is on central government land, the settlement has no possibility under current law for gaining legal recognition. As such, the underlying structural origins that created spatial stigma, as well as many of the underlying material conditions, have not changed substantially in the community. In addition, by applying the concept of spatial stigma to a new

context, the primary contribution of this paper is to public health and social theory. As such, the age of data is less salient and does not detract from the theoretical contributions of this work. In addition, since the initial data collection did not explicitly focus on spatial stigma, we recommend that future studies in KB should investigate differences in gender and socioeconomic status that have been recognized as underlying the association between spatial stigma and health in other contexts [34], as well as the intersection of ethno-religious identities and caste status that are relevant in this locality.

## Conclusion

By addressing the "elephant in the room" [12], our empirical analysis of the mental health effects of spatial stigma reveals key mechanisms through which the subjective experiences of living in a disadvantaged neighborhood can impact the mental health of its residents. Previous research in KB, reflecting the primary thrust of research on place and health, has revealed important associations between the neighborhood's physical conditions and social composition and its residents' risk for probable CMD [24]. The present focus on spatial stigma further deepens our understandings of the health effects of living in disadvantaged neighborhoods. Furthermore, our emphasis on the structural causes and the enactment of stigma-power by government officials allows us to draw out these connections without attributing the stigma's source to residents' behaviors, which can contribute to further stigmatization of disadvantage neighborhoods like KB. This study provides empirical evidence for the mental health impacts of spatial stigma and contributes understandings of the health effects of living in disadvantaged, stigmatized neighborhoods.

## Acknowledgments

PUKAR's barefoot researchers, many of whom live in the Kaula Bandar community, helped to facilitate the recruitment of study participants for the qualitative interviews and focus group discussions. Kalyani Monteiro Jayasankar helped with the implementation of focus group discussions. Devorah Klein Lev-Tov helped transcribe hours of interviews. We are grateful to the residents of KB for opening their doors to us and sharing their life stories.

## Author Contributions

**Conceptualization:** Saloni Dev, Liza Weinstein.

**Data curation:** Saloni Dev.

**Formal analysis:** Saloni Dev, Jasper Duval, Amith Galivanche, Liza Weinstein.

**Funding acquisition:** Ramnath Subbaraman, Liza Weinstein.

**Investigation:** Tejal Shitole, Kiran Sawant, Shrutika Shitole, Anita Patil-Deshmukh.

**Methodology:** Saloni Dev.

**Project administration:** Saloni Dev, Tejal Shitole, Kiran Sawant, Shrutika Shitole, Anita Patil-Deshmukh.

**Resources:** Anita Patil-Deshmukh, Liza Weinstein.

**Software:** Saloni Dev.

**Supervision:** Saloni Dev, Anita Patil-Deshmukh, Ramnath Subbaraman, Liza Weinstein.

**Validation:** Saloni Dev, Ramnath Subbaraman, Liza Weinstein.

**Visualization:** Saloni Dev, Ramnath Subbaraman, Liza Weinstein.

**Writing – original draft:** Saloni Dev, Liza Weinstein.

**Writing – review & editing:** Saloni Dev, Alisa Lincoln, Ramnath Subbaraman, Liza Weinstein.

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
