## [Decision Letter · Decision Letter 0]

8 Dec 2022

PGPH-D-22-01320

Spatializing Stigma-Power Mental Health Impacts of Spatial Stigma in a Legally-Excluded Settlement in Mumbai

Dear Dr. Dev,

Thank you for submitting your manuscript to PLOS Global Public Health. After careful consideration, we feel that it has merit but does not fully meet PLOS Global Public Health’s publication criteria as it currently stands. Therefore, we invite you to submit a revised version of the manuscript that addresses the points raised during the review process.

Authors need to extensively edit the paper to improve wider readability to a global health audience, address inconsistencies and gaps in aims and methods, and perhaps reflect on and address the particular issue of the slant/gaze ("deficit language" as the reviewer points out). Please also see other major comments of the reviewer which need to be addressed. 

We look forward to receiving your revised manuscript.

Kind regards,

Prashanth Nuggehalli Srinivas, MBBS, MPH, PhD

Academic Editor

Journal Requirements:

1. In the ethics statement in the Methods, you have specified that verbal consent was obtained. Please provide additional details regarding how this consent was documented and witnessed, and state whether this was approved by the IRB.

If you did not receive any funding for this study, please simply state: “The authors received no specific funding for this work.

3. Please provide separate figure files in .tif or .eps format only and remove any figures embedded in your manuscript file. Please also ensure that all files are under our size limit of 10MB.

4. We notice that your supplementary table is included in the manuscript file. Please remove them and upload them with the file type 'Supporting Information'. Please ensure that each Supporting Information file has a legend listed in the manuscript after the references list.

5. In the online submission form, you indicated that "Given confidentiality and privacy concerns, de-identified qualitative data will be available from authors upon request". All PLOS journals now require all data underlying the findings described in their manuscript to be freely available to other researchers, either 1. In a public repository, 2. Within the manuscript itself, or 3. Uploaded as supplementary information.

6. Images 1 and 2: please (a) provide a direct link to the base layer of the map (i.e., the country or region border shape) and ensure this is also included in the figure legend; and (b) provide a link to the terms of use / license information for the base layer image or shapefile. We cannot publish proprietary or copyrighted maps (e.g. Google Maps, Mapquest) and the terms of use for your map base layer must be compatible with our CC-BY 4.0 license. 

Additional Editor Comments (if provided):

Reviewers' comments:

Reviewer's Responses to Questions

**Comments to the Author**

1. Does this manuscript meet PLOS Global Public Health’s publication criteria? Is the manuscript technically sound, and do the data support the conclusions? The manuscript must describe methodologically and ethically rigorous research with conclusions that are appropriately drawn based on the data presented.

Reviewer #1: Yes

2. Has the statistical analysis been performed appropriately and rigorously?

Reviewer #1: Yes

3. Have the authors made all data underlying the findings in their manuscript fully available (please refer to the Data Availability Statement at the start of the manuscript PDF file)?

Reviewer #1: Yes

4. Is the manuscript presented in an intelligible fashion and written in standard English?

Reviewer #1: Yes

5. Review Comments to the Author

Reviewer #1: General comments

1. Thanks – this is a fascinating and important paper and the abstract is brisk and clear.

2. I think for a global public health paper the paper could generally lean into a more concise and simple phraseology – the social science influence is evident in lengthy sentences and poly syllabic vocabulary which in some instances could be phrased more simply and accessibly. This is important in a global health journal where English is not a first language for many readers. For example a sentence like this takes quite slow and careful reading to elicit the intended meaning: Most importantly, we found that structural factors independently influenced disinvestment in infrastructural resources for KB as a form of exercise of stigma power to achieve the goals of eviction and clearing up the land.

3. Researcher stance and authorship– can you please clarify the intersectional identity of authors and how they contributed. In particular, as the first three and two senior authors are affiliated with a HIC institutions and not with the Mumbai based research organisation, how does this paper contribute to reversing of the gross inequities in authorship related to LMIC and HIC residents of authors, which have been perpetuated for many decades ie. where the people with local knowledge and residence are assigned low prestige authorship positions. The asymmetry in authorship is stark and mirrors the power and privilege of authorship for high income country residing authors, an important concern, especially in a paper like this where the focus is on local knowledge/ context/ location and power. A discussion of authorship stance would also be important to include in the methods section to acknowledge how intersectional identity influences research design and reporting.

Introduction

4. Literature is well summarised and the research gap clearly identified. Given the Dhaka study on spatial sitgma is reported as the only one in South Asia it would be useful to get a one sentence summary of that.

5. Can you give further data or evidence showing that this informal urban setting or residents in slums generally have higher rates of CMDs than other settings where people with low income reside in India/ Mumbai?

6. Given the centrality of the Pearce framework on spatial stigma which it seems was developed in a high income context, it would be useful to consider what contextual factors are at play and different in the setting you describe - eg in India/ South Asia, especially as ‘slums’ have more legally contested locations which adds another layer of structural disadvantage.

7. Aims - I think this is your statement of aim – can you say more simply? In this paper, we draw on the valuable, but still limited research on spatial stigma and health to investigate the extent to which stigmatized perceptions of a neighborhood may help explain the high prevalence of probable common mental health disorders (CMD) in a non notified slum (i.e., settlement lacking legal recognition by the government) in Mumbai, India.

Or is this other statement also apparently summarising the study aim : In the analysis that follows, we use this integrated conceptual framework as a starting point for interrogating pathways between spatial stigma and mental health in a slum settlement in Mumbai, India, that has been previously shown to have a very high prevalence of CMDs.(25) Of note, this is also an inquiry into the applicability of Pearce’s (38) conceptual framework in the slum context in India, and the inter-relationships between these pathways (Pearce (38) acknowledged that these pathways are non-mutually exclusive but did not articulate the relationships between them).

It would be preferable to state study objectives in one place and state things more directly

EG We will use the Pearce framework to interrogate pathways between spatial stigma and poor mental health and also examine how applicable and relevant the Pearce framework is in a slum setting in India.

8. History of study setting – wonderful to see this written with clear pointing to colonisation and historical health determinants, refreshing and hopefully sets the standard in political economy of health and place.

9. Appreciate the rationale for relevance of data collected over a decade ago.

10. Table 1 – notable is the absence of information on caste identity as a major health determinant in India. Did you gather this information and can it be added? If not can you please comment on caste and how it may have interacted in findings of this study. Also, perhaps you can give a small rationale for your definition of North and South Indians and relevance of those identities vs Maharashtrians.

11. Methods-

See comment earlier about stance of authors

12. Results

Nice to get social identity of people who participate in verbatim quotes – thank you.

13. Apart from the data around self stigma, the authors present few results linking mental distress and structural determinants directly. I think it is reasonable to assume that the textured descriptions of social stigma and how it operates lead to mental distress but dering if you have any data that draws this connection more explicitly.

14. This study presented a largely deficit focused summary of the mental health stigma experiences and of discrimination and exclusion linked to residence in this slum. I appreciated the small section in results which identified a positive neighbourhood identity of community cohesion and lack of communitarian violence during the early 90’s. There is a risk in presenting such a paper that it further entrenches negative assessments of this and similar slums with a largely negative picture. I wonder if you have any further data that can also examine and nuance your data by also describing community assets and examples of mental health resilience or strength that may also be linked to this spatial location. This would add some nuance and I believe would not take away from the over-arching study question.

6. PLOS authors have the option to publish the peer review history of their article (what does this mean?). If published, this will include your full peer review and any attached files.

**Do you want your identity to be public for this peer review?** For information about this choice, including consent withdrawal, please see our Privacy Policy.

Reviewer #1: **Yes: **Kaaren Mathias

---

## [Decision Letter · Decision Letter 1]

16 May 2023

PGPH-D-22-01320R1

Spatializing Stigma-Power Mental Health Impacts of Spatial Stigma in a Legally-Excluded Settlement in Mumbai

Dear Dr. Dev,

Thank you for submitting your manuscript to PLOS Global Public Health. After careful consideration, we feel that it has merit but does not fully meet PLOS Global Public Health’s publication criteria as it currently stands. Therefore, we invite you to submit a revised version of the manuscript that addresses the points raised during the review process.

Please address some of the additional points raised by reviewer 2 below. 

We look forward to receiving your revised manuscript.

Kind regards,

Prashanth Nuggehalli Srinivas, MBBS, MPH, PhD

Academic Editor

Journal Requirements:

1. In the ethics statement in the Methods, you have specified that verbal consent was obtained. Please provide additional details regarding how this consent was documented and witnessed, and state whether this was approved by the IRB.

Additional Editor Comments (if provided):

Reviewers' comments:

Reviewer's Responses to Questions

**Comments to the Author**

1. If the authors have adequately addressed your comments raised in a previous round of review and you feel that this manuscript is now acceptable for publication, you may indicate that here to bypass the “Comments to the Author” section, enter your conflict of interest statement in the “Confidential to Editor” section, and submit your "Accept" recommendation.

Reviewer #1: All comments have been addressed

Reviewer #2: (No Response)

2. Does this manuscript meet PLOS Global Public Health’s publication criteria? Is the manuscript technically sound, and do the data support the conclusions? The manuscript must describe methodologically and ethically rigorous research with conclusions that are appropriately drawn based on the data presented.

Reviewer #1: Yes

Reviewer #2: Yes

3. Has the statistical analysis been performed appropriately and rigorously?

Reviewer #1: N/A

Reviewer #2: N/A

4. Have the authors made all data underlying the findings in their manuscript fully available (please refer to the Data Availability Statement at the start of the manuscript PDF file)?

Reviewer #1: No

Reviewer #2: Yes

5. Is the manuscript presented in an intelligible fashion and written in standard English?

Reviewer #1: Yes

Reviewer #2: Yes

6. Review Comments to the Author

Reviewer #1: I have no further comments.

Reviewer #2: Title: Spatializing Stigma-Power: Mental Health Impacts of Spatial Stigma in a Legally-Excluded Settlement in Mumbai, India

Major Issues

• The authors need to further explain how their findings reflects the socio-ecological framework they have used in this study. Also the authors must show the link between the socio-ecological framework and Pearce Conceptual Model in relation to the five pathways identified in the study.

Minor Issues

• The authors should add a map before the Methods section to depict the social process that has led to some form of change in the social environment.

• The social processes of spatial stigma should be discussed. This is because; spatial stigma is a new social construction of reality within the domain of public health research.

• The sub-section “Conclusion” should be replaced with “Discussion/Conclusion” or this should be separated.

• The authors should consider adding “India” at the end of their title after “Mumbai”.

• GENERAL OBSEVATION

• The manuscript should be double-checked prior to publication to address any form of spelling mistakes on error.

• The study is quite interesting and its replication in other settings will further assert the concepts of “Stigma Power” and Spatial Stigma.

7. PLOS authors have the option to publish the peer review history of their article (what does this mean?). If published, this will include your full peer review and any attached files.

**Do you want your identity to be public for this peer review?** For information about this choice, including consent withdrawal, please see our Privacy Policy.

Reviewer #1: **Yes: **Kaaren Mathias - Senior Lecturer, University of Canterbury, NZ and Senior Advisory, Burans, Emmanuel Health Association, India

Reviewer #2: **Yes: **Saheed Akinmayowa Lawal

---

## [Editor Report · Decision Letter 2]

12 Jun 2023

Spatializing Stigma-Power Mental Health Impacts of Spatial Stigma in a Legally-Excluded Settlement in Mumbai

PGPH-D-22-01320R2

Dear Dr. Dev,

We are pleased to inform you that your manuscript 'Spatializing Stigma-Power Mental Health Impacts of Spatial Stigma in a Legally-Excluded Settlement in Mumbai' has been provisionally accepted for publication in PLOS Global Public Health.

Best regards,

Prashanth Nuggehalli Srinivas, MBBS, MPH, PhD

Academic Editor